# In Vivo Ultrafast Doppler Imaging Combined with Confocal Microscopy and Behavioral Approaches to Gain Insight into the Central Expression of Peripheral Neuropathy in Trembler-J Mice

**DOI:** 10.3390/biology12101324

**Published:** 2023-10-10

**Authors:** Mariana Martínez Barreiro, Lucia Vázquez Alberdi, Lucila De León, Guadalupe Avellanal, Andrea Duarte, Maximiliano Anzibar Fialho, Jérôme Baranger, Miguel Calero, Nicolás Rubido, Mickael Tanter, Carlos Negreira, Javier Brum, Juan Pablo Damián, Alejandra Kun

**Affiliations:** 1Laboratorio de Biología Celular del Sistema Nervioso Periférico, Departamento de Proteínas y Ácidos Nucleicos, Instituto de Investigaciones Biológicas Clemente Estable, Montevideo 11600, Uruguay; mmartinezb@iibce.edu.uy (M.M.B.); lvazquez@iibce.edu.uy (L.V.A.); aduarte@iibce.edu.uy (A.D.); 2Laboratorio de Acústica Ultrasonora, Instituto de Física, Facultad de Ciencias, Universidad de la República, Montevideo 11400, Uruguay; manzibar@fisica.edu.uy (M.A.F.); carlosn@fisica.edu.uy (C.N.); jbrum@fisica.edu.uy (J.B.); 3Departamento de Biociencias Veterinarias, Facultad de Veterinaria, Universidad de la República, Montevideo 13000, Uruguay; luciladeleonn@gmail.com (L.D.L.); galloguada@gmail.com (G.A.); jpablodamian@gmail.com (J.P.D.); 4Física No Lineal, Instituto de Física de Facultad de Ciencias, Universidad de la República, Montevideo 11400, Uruguay; nicolas.rubidoobrer@abdn.ac.uk; 5Physics for Medicine Paris, Inserm U1273, ESPCI Paris, PSL University, CNRS UMR 8063, 75012 Paris, France; jerome.baranger@espci.fr (J.B.); mickael.tanter@espci.fr (M.T.); 6Unidad de Encefalopatías Espongiformes, UFIEC, CIBERNED, Instituto de Salud Carlos III, 28029 Madrid, Spain; mcalero@isciii.es; 7Queen Sofia Foundation—Alzheimer Center, CIEN Foundation, 28031 Madrid, Spain; 8Institute for Complex Systems and Mathematical Biology, University of Aberdeen, King’s College, Aberdeen AB24 3UE, UK; 9Sección Bioquímica, Facultad de Ciencias, Universidad de la República, Montevideo 11400, Uruguay

**Keywords:** Ultrafast Power Doppler, Scanning Laser Confocal Microscopy, behavioral tests, Trembler-J, CMT1E, anxiety, hippocampi

## Abstract

**Simple Summary:**

In this work, we explore the central compromise in TrJ/+ mice, a model for the peripheral neuropathy Charcot-Marie-Tooth, using three different approximations: Ultrafast Doppler, Confocal Microscopy, and behavioral tests, exposing alterations in the brain vasculature, as well as an anxiety-like behavior. Hemodynamic changes recorded in vivo, associated with vascular volume modulation, together with behavioral alterations in the TrJ/+ model, account for a functional-structural-behavioral profile that demonstrates vascular/central involvement of the disease.

**Abstract:**

The main human hereditary peripheral neuropathy (Charcot-Marie-Tooth, CMT), manifests in progressive sensory and motor deficits. Mutations in the compact myelin protein gene pmp22 cause more than 50% of all CMTs. CMT1E is a subtype of CMT1 myelinopathy carrying micro-mutations in pmp22. The Trembler-J mice have a spontaneous mutation in pmp22 identical to that present in CMT1E human patients. PMP22 is mainly (but not exclusively) expressed in Schwann cells. Some studies have found the presence of pmp22 together with some anomalies in the CNS of CMT patients. Recently, we identified the presence of higher hippocampal pmp22 expression and elevated levels of anxious behavior in TrJ/+ compared to those observed in wt. In the present paper, we delve deeper into the central expression of the neuropathy modeled in Trembler-J analyzing in vivo the cerebrovascular component by Ultrafast Doppler, exploring the vascular structure by scanning laser confocal microscopy, and analyzing the behavioral profile by anxiety and motor difficulty tests. We have found that TrJ/+ hippocampi have increased blood flow and a higher vessel volume compared with the wild type. Together with this, we found an anxiety-like profile in TrJ/+ and the motor difficulties described earlier. We demonstrate that there are specific cerebrovascular hemodynamics associated with a vascular structure and anxious behavior associated with the TrJ/+ clinical phenotype, a model of the human CMT1E disease.

## 1. Introduction

The most prevalent human Peripheral Nervous System (PNS) disorder is the Charcot-Marie-Tooth disease (CMT): a diverse group of hereditary, chronic, progressive, sensory/motor peripheral neuropathies caused by monogenic mutations [1,2,3]. Different genes, in neurons and Schwann cells, are involved in these disorders, showing more than 1000 different mutations in 80 genes [4]. The main involved genes are *pmp22* (coding to Peripheral Myelin Protein 22, PMP22), gjb1 (coding to gap junction protein B1, Connexin-32, Cx-32), MPZ (coding to Myelin Protein Zero, MPZ), litaf (coding to Lipopolysaccharide Induced TNF Factor, LITAF), mfn2 (coding to Mitofusin 2, MFN2), egr2 (coding to Early Growth Response 2, EGR2), and sh3tc2 (coding to SH3 Domain and Tetratricopeptide Repeat-Containing Protein 2, SH3TC2) [5,6,7,8]. With a few exceptions, most mutations observed in PMP22 show autosomal dominant heritability, autosomal recessive and X-linked inheritance being less common [5]. Classically, two different types of CMT are distinguished: Type I CMTs generally considered as schwannopathies, with mutations in CS’s genes, and type II CMTs involving axonopathies, having neuronal gene mutations. The *pmp22* mutations cause about 70% of all CMT type 1 (CMT1) [5,6]. *Pmp22* is a highly conserved 40 kb/6 exons gene, belonging to the growth arrest-specific genes characterized by Shneider et al. in 1988 [9]. It was early characterized in NIH3T3 murine fibroblasts, due to a noticeable increased transcription during cell cycle arrest [10]. Later, two of its transcripts (CD25 and SR13) were identified in rat sciatic nerve, associated with the Schwann cell and myelin, suggesting a plausible structure of four transmembrane domains [11,12] and a repeatedly confirmed N-glycosylation site at Asparagine 41, at the first extracellular domain [11,13,14,15], in the finally translated protein. Analogous transcripts were also found in mice and humans [16,17]. Known mutations in PMP22 include 44 single base substitutions, 14 deletions, two insertions, one reciprocal translocation, several excision sites, and some single base substitutions in exon 1A and in the 3’ end UTR. All of them are integrated in the CMT1E subtype [4,5]. The PMP22 results in a small (22 kDa) hydrophobic tetraspan claudin, highly expressed in the peripheral nervous system myelin [14,18,19,20,21,22]. Among them, mutations affecting the pmp22 (encoding peripheral-myelin-protein-22, PMP22) cause about 70% of all CMT type 1 (CMT1), called myelinopathies [5,6]. CMT1E is a subtype of CMT1 carrying micromutations in the pmp22 [4,5]. The PMP22 has historically been found only in the peripheral nervous system, specifically bound to compact myelin [14,18,19,20], it is a glycosylated claudin with functions in the regulation of cell growth and differentiation [19,21,22]. However, some studies have also found the presence of PMP22 in the Central Nervous System (CNS). The pmp22 transcript has been found in whole brain extracts [11,12,23], in neurons of cranial and spinal nerves [24] and in the CNS [25]. In addition, some CNS implications have been found in patients with CMT. In cases of familial CMT1, lesions have been found in the cerebral white matter [26]. A case of CNS demyelination has been described in a patient with CMT1a mimicking multiple sclerosis [27]. More recently, functional reorganization in multiple large-scale networks has been found in patients with CMT1 [28].

The elucidation of PMP22’s roles and functions has contributed to the understanding of CMT1E pathogenesis [4,5,14]. Trembler (TrJ)and Trembler-J (TrJ/+) murine models have elucidated some of the clinical phenotypes caused by defective processing of mutant PMP22 and altered intracellular trafficking [29,30]. The TrJ/+, in particular, have a spontaneous point mutation in the pmp22 gene (T1703C) which results in an L16P change, affecting the first transmembrane domain of PMP22, preventing its insertion and generating intracellular aggregates, with a toxic gain of function, hypomyelination, and axonal degeneration [15,29,31,32,33,34,35,36,37]. This neurodegenerative phenotype presents different levels of severity depending on gene dosage: while TrJ/+ heterozygotes are viable, recessive homozygotes (TrJ/TrJ) die before weaning [38,39]. The main clinical manifestations are spastic paralysis and generalized tremor. TrJ is a model of high biological fidelity to the human condition, as the same mutation is found in the homologous gene, of a CMT1E lineage [14,18,29,40]. The underlying cellular homeostasis in the TrJ model is shifted towards a progressive deterioration in the efficiency of nerve fiber maintenance. Aggregation of the mutated PMP22 protein saturates cellular detoxification pathways, generating a gain of toxic function, increased oxidative stress, decreased antioxidant response, and mitochondrial alteration [41,42,43,44,45,46]. This is a particularly critical context in the SNP where PMP22 shows its maximum expression. However, the biological consequences of the micromutation generate a complex phenotype which, as we have shown, also manifests in the CNS [47,48]. In addition, its presence in other cell types, and at the nuclear level, augurs other unelucidated roles with more systemic characteristics in the expression of the TrJ phenotype. Recently, our group reported for the first time the presence of pmp22 in the TrJ hippocampus, together with a behavioral profile of the anxious type [47].

The diversity of genes and mutations causing CMT conditions produces phenotypes of varying severity, both in the onset and progression of the disease. There are no specific therapies for any form of CMT, the attenuation of clinical symptomatology being the most common application (i.e., vitamin supplements, pes cavus surgery) [49,50]. However, at present, although still in the testing stages in animal models, advances in virus-mediated gene therapies, the identification of sensitive molecular targets, and various pharmacological approaches aimed at improving clearance of protein aggresomes seem to be in the direction of a personalized medicine of the CMTs [51,52].

Cerebrovascular physiology is a key element in the understanding of brain health. Neurovascular biology underpins and provides insight into relevant aspects of cognitive and behavioral function, aging, or neurodegenerative progression [53,54]. Vascular alterations of the brain have been observed in the development of neurodegenerative diseases, in animal models and in humans [55,56]. Some crossectional and longitudinal clinical studies reveal that impaired blood flow is a common and early indicator of Alzheimer’s disease (AD), postulating that it may even precede the onset of proteinopathy in its symptomatic stage, affecting brain perfusion and connectivity [53,57,58]. However, the elucidation of the role of the vascular component in neurodegenerative homeostasis has not yet been resolved for most nervous system disorders. Vascular dysfunctions impact cellular oxygen pathways, including glucose metabolism, oxidative phosphorylation, and mitochondrial and cellular homeostasis of neurons and glia [59,60,61].

The vasculature of the brain is fundamental for its proper functioning, providing oxygen and nutrients, regulating immune trafficking and clearing pathogenic proteins [54]. Due to the large number of functionally distinct brain regions with different nutritional needs and the high energy demand of the brain, supplying the right amount of oxygen to each region is a major logistical challenge. Because of this, disturbances in blood flow can be detrimental to the CNS’s healthy functioning, which plays an important role in neurodegenerative diseases. Blood–brain barrier permeability and blood flow disturbance have been detected in initial AD disease, together with brain infarcts, arterial lipid deposits, and arterial wall thickening [54,62]. A clear relationship exists between brain neurovasculature and brain health, with cerebrovascular dysfunction being a cause/effect phenomenon associated with neurodegenerative diseases. In CMT, the study of the vascular compromise has been poorly signaled, despite having been originally described as accompanied by vasomotor abnormalities [63,64]. Recently, we have reported the presence of PMP22 protein in the TrJ model at the hippocampal level. Consistently, an anxious behavior seems to involve the hippocampal domines as a component of the TrJ clinical phenotype [47].

New technologies and tools have made it possible to analyze the brain from other perspectives. Recently, our group started working on a new image-driven modality, Ultrafast Doppler (µDoppler), a powerful tool for in vivo imaging of cerebral blood flow. This technique allows us to observe the cerebral blood volume (CBV) with high sensitivity. Precisely, using this technique in association with Scanning Laser Confocal Microscopy (SLCM), we developed a method for quantifying the blood flow and its corresponding vascular structure, differentiating the CBV in quartiles according to the different vessel sizes, and finding that the CBV and the vascular structure vary with age [48].

In this study, we evaluate the in vivo cerebral blood flow and vascular structure distribution in TrJ mice by µDoppler and SLCM. In addition, in order to understand the TrJ phenotype as a whole, we explore the associated behavioral component in anxiety tests.

## 2. Materials and Methods

### 2.1. Animals

The local ethics committee approved all the experiments and procedures (Comisión de Ética en el Uso de Animales (CEUA), Instituto de Investigaciones Biológicas Clemente Estable (IIBCE), Uruguay, protocol number: 002a/10/2020). The regulations and guidelines were followed strictly in all the experiments (Uruguayan Law number 18611, accessed on 2 October 2019, link to the law: https://www.impo.com.uy/bases/leyes/18611-2009/8).The endogamic mice strain B6.S2-Pmp22^Tr-J/j^m rJ/+ and the wild type for Pmp22, +/+ were acquired from Jackson Laboratories. Both strains were bred in the IIBCE animal facility and raised under controlled conditions, with free access to water and food, with dark/light cycle (12 h/12 h), at 21 ± 3 °C. From an early age, the phenotype of TrJ/+ mice is distinguished from +/+ by the suspending tail test, as was reported by Rosso et al. [65]. Three-month-old male animals were used for the experiments, following the distribution shown in Table 1.

### 2.2. µDoppler Images Acquisition

For µDoppler acquisition, the mouse was anesthetized with 120 mg/kg ketamine (Ventanarcol, Koing do Brasil Ltda., São Paulo, Brazil) and 16 mg/kg xylazine (Xylased*2, Vetcross) diluted in 300 µL of saline solution. Then, its head was shaved to avoid interference with the ultrasound signal caused by the air trapped inside the fur. A 128-element, 15 MHz ultrasound probe driven by Verasonics Vantage System was used for µDoppler imaging. To this end, each mouse was placed in a customized stereotaxic frame that allowed alignment of the ultrasound probe with the coronal plane of the brain. Each µDoppler image was generated by averaging 350 frames using a four-angle compound sequence and applying clutter filtering based on singular value decomposition (SVD) (Figure 1b). The cut-off values used in the SVD clutter filter were selected based on achieving on the best signal-to-noise ratio. Further information regarding this experimental procedure can be found in [48].

### 2.3. µDoppler Images Analysis

For the quantification of µDoppler images, the Matlab software was used. A program was generated to select the hippocampus and cortex section and separate the intensity levels of the pixels into quartiles, corresponding to the different structures of the vessels: big arteries and vein, smaller arteries and vein, arteriole, and capillary-venules (Figure 1b). The number of pixels in the whole section was used to normalize the data.

### 2.4. Brain Processing for Vibratome Sectioning

Mouse’s brain was dissected immediately after cervical dislocation euthanasia and fixed by immersion in 4% PFA in PHEM buffer (60 mM PIPES, 25 mM HEPES, 10 mM EGTA, 2 mM MgCl_2_, adjusted to pH 7.2–7.4 with KOH pellets) at 4 °C for 24 h. After that, the brain was washed in large volumes of PHEM buffer, each for 5 min 6 times, to eliminate excess fixatives. The brain was then embedded in a block with a mixture of 0.5% gelatin, 30% bovine serum albumin, and 1% glutaraldehyde (final concentration). Vibratome sections of 60 µm thickness were obtained in a Leica, VT 10000S vibratome. For vessel visualization, brain sections containing the hippocampal head regions were stained as previously described using Isolectin GS-IB4 Alexa Fluor 488 conjugate (Cat#: I21411, ThermoFisher Scientific, Waltham, MA, USA) in 1:100 concentration (Figure 1c) [48].

### 2.5. Scanning Laser Confocal Microscopy

For cerebrovascular imaging, the Zeiss LSM 800 confocal microscope was used. Applying the same voltage and photomultiplier conditions and performing a 10-plane scan on the *Z*-axis, images of the same SLCM section were obtained. In addition, the tail scan mode was used to compose the images of the coronal section of the brain (Figure 1c).

### 2.6. Confocal Image Analysis

For the quantification, the confocal images of two consecutive brain slices were used, to form a thickness similar to the µDoppler image. Using ImageJ software, each hippocampus and cortex section was selected and created a binary image, using the automatic threshold function. The 3D counter plug-in was used to analyze the vascular volumes and the number of vessels. In order to normalize the results, the volume of the hippocampus or cortex section (total volume) was used, defining the value of the vessel volume fraction (*VVF*):


$$\mathrm{VVF}=\frac{\mathrm{Vessel}\mathrm{Volume}}{\mathrm{Total}\mathrm{Volume}}\times 100$$


The *VVF* distribution was divided into quartiles, sectioning the whole vessels into four groups where the sum of volumes in each group corresponds to 25% of the sum of all vessel’s volumes, in decreasing order. Each of them corresponds to Q1 (large arteries and veins), Q2 (smaller arteries and veins), Q3 (arterioles and venules), and Q4 (capillaries and venules).

### 2.7. Behavioral Tests

The Open Field Test and the Elevated Plus Maze were used as behavioral tests to assess anxiety, while the Rotarod was used to assess motor behavior (Figure 1a). As reported by Damián et al. (2021) [47], the animals were acclimatized for at least 2 h before performing each of the tests. All the tests were carried out (on different days) in a room at a controlled temperature (20 ± 2 °C). After each of the mice went through each test, the apparatus was sanitized using 70% alcohol.

#### 2.7.1. Open Field Test

For the Open Field Test, a plexiglass box with the following dimensions 30 × 35 × 40 cm was used. Video recordings were made during the 10 min that each test lasted. The behaviors evaluated during the Open Field Test were the number of rearings, grooming, freezing, fecal boli, and head shakes, as well as the time dedicated to grooming and freezing [47,66].

#### 2.7.2. Elevated Plus Maze Test

The apparatus for the Elevated Plus Maze Test (length of each arm 30 × 5 cm) was the same as the one previously used by Damián et al. [47,66]. During the test that lasted five minutes, the number of entries in open and closed arms and total entries were recorded, and in addition, the number of grooming, rearing, fecal boli, and head shakes were also recorded [47,66].

#### 2.7.3. Rotarod Test

For the Rotarod, a cylindrical motorized platform (5 cm in length × 5 cm in diameter), which rotates at different velocities, was used. Mice are placed above the cylinder and the velocities are augmented at 15 s, until reaching the 5 different velocities. The time of permanence in the platform for each of the speeds was scored.

### 2.8. Statistical Analysis

Normality was evaluated using the Shapiro–Wilk test. Behavioral parameters for TrJ/+ and +/+ phenotypes were compared using Student’s test when normally distributed while the Mann–Whitney U test was used for non-normal distributed parameters. Different quartiles within the same phenotype were compared using the one-way ANOVA test with the Bonferroni test for multiple comparisons as post hoc for normal distributions, and the Friedman test for non-normal distributions. For confocal microscopy quantification, because of the great variability, the data were analyzed in function of each quartile separately.

## 3. Results

### 3.1. µDoppler Images Quantification

The different quartiles showed significant differences for each genotype, both in the hippocampus and in the cortex (*p* < 0.01, for all comparisons, Figure 2c,d and Figure 3c,d).

In the hippocampus, when comparing the mean quartile value between both genotypes, the TrJ/+ mice show a significantly higher number of decibels for each quartile (*p* < 0.001, for all comparisons, Figure 4a), while in the cortex there were no significant differences between genotypes (Figure 4b).

### 3.2. Confocal Microscopy Vascular Visualization

The 3D Object counter plug-in and posterior data analysis showed significant differences between each quartile for each genotype, (Figure 5), both in the hippocampus and cortex section (*p* < 0.0001 for all comparisons).

Figure 6 shows the comparison of the vessels between different genotypes using confocal microscopy in the hippocampus and cortex section. In the hippocampus, the TrJ/+ mice showed higher mean VVF values for each quartile, compared to Wt (+/+ vs. TrJ/+: *p* < 0.0001 for all comparisons) (Figure 6a). In the cortex section, no significant differences were found between both genotypes for Q1 and Q2, but for Q3 and Q4 TrJ/+ mice showed higher mean VVF values compared to Wt (+/+ vs. TrJ/+: [Q3]: *p* < 0.0001; [Q4]: *p* = 0.0160) (Figure 6b).

### 3.3. Behavioral Tests

#### 3.3.1. Elevated Plus Maze

In the Elevated Plus Maze test, TrJ/+ mice presented lower frequency of closed-arm entries (*p* = 0.0065) (Figure 7a), lower total entries (*p* = 0.0380) (Figure 7b), lower rearing frequency (*p* < 0.0001) (Figure 7c), and greater defecation frequency (*p* < 0.0001) (Figure 7d) than Wt mice.

#### 3.3.2. Open Field Test and Rotarod

In the Open Field Test, compared with Wt mice, TrJ mice presented more frequency and time spent freezing (*p* < 0.001) (Figure 8a,b) more frequency and time spent grooming (*p* < 0.001) (Figure 8c,d), and lower frequency of rearing (*p* < 0.0001) (Figure 8e), headshakes (*p* < 0.0001) (Figure 8f), and defecation (*p* < 0.0001) (Figure 8g).

In the Rotarod test, TrJ/+ mice presented lower time of permanence than wt mice (*p* < 0.0001) (Figure 8h).

## 4. Discussion

In the present work, we address unexplored aspects of the CMT1E central expression, modeled in TrJ/+ mice, through a functional, structural, and behavioral analysis of the neurodegenerative phenotype.

We report here, for the first time, an increase in the cerebral hippocampal perfusion of TrJ/+ mice, compared to that observed under the same conditions in the hippocampi of +/+ mice (Figure 4a). In addition, the volumetry of the cerebrovascular network was further analyzed by SLCM, showing that hippocampal TrJ/+ vessel volume was larger than in the +/+ brain for mice of the same age (Figure 6a). The finding was made possible by non-invasive in vivo µDoppler imaging using erythrocytes as ultrasound diffusing elements and subsequent combination with confocal mosaic imaging of the Isolectin IB4-labeled vascular network (post mortem). The segmentation of the data distribution of both imaging into quartiles, represents with reasonable fidelity the structural functionality of the vascular network (Q1: large arteries and veins; Q2: smaller arteries and veins; Q3: arterioles and venules; Q4: capillaries and venules). This tool allowed us a quantitative assessment in the characterization of vascular aging in wt mice, which we have recently reported [48]. This combined approach enhances and verifies the functional findings with a higher-resolution description of the vascular structure.

In the present study, we verified that the anxious-like behavior, which we had previously described in 5-month-old TrJ/+ mice [47], is already present at an earlier age. Thus, the hippocampal activity in TrJ/+ may require hyperperfusion, sustained by a greater volume of vessels rather than an increase in their number. Interestingly, we found no significant differences between genotypes when analyzing the number of cortex and hippocampal normalized vessels (Appendix A). These data suggest that in TrJ/+, hyperperfusion is accompanied by a sustained expansion or dilation of blood vessel volume in the hippocampal region.

No differences in the suprahippocampal cortex perfusion were observed between genotypes (Figure 4b). However, although the Q1 and Q2 vessel volumes did not show any differences, a significant change was observed in vessel volumes mainly in Q3 and, with less significance, in Q4 (Figure 6b). The absence of differences in the number of vessels (Appendix A), suggests that the normal cortical perfusion is supported mainly by a dilation of arterioles and (partially) capillaries-venules, contained in quartiles 3 and 4 of TrJ cortex.

Lastly, a different behavioral profile was found in TrJ/+ mice, showing an anxiety-type behavior, as seen by higher frequency of and time spent freezing and grooming, and higher frequency of rearing and defecation. Additionally, TrJ/+ mice showed the notorious presence of headshakes, which could correspond to a central compromise [67]. Other behavioral tests confirm the motor difficulty known to be present in TrJ/+ mice, as the lower permanence in Rotarod and lower total entries in EPM. Although the EPM test shows responses to stress, in this case, the lower frequency of entries in closed arms of TrJ/+ mice can be explained by the motor difficulties present. These results reaffirm that the behavioral profile of TrJ/+ mice differs to +/+, evidencing an anxious-type profile, as reported previously by Damian et al. [47]. It is interesting to highlight that although the mice used in this work were younger than the ones used in Damian et al. [47], the profile is almost identical, which allows us to speculate that the behavioral profile of TrJ/+ is characteristic of the pathologic condition, and is not affected in relationship with the age. Additionally, this also reinforces that the vascular changes observed in this study are contrasted with behavioral variables associated specifically with the brain areas, for example, the hippocampus.

Changes in the cerebral vasculature have been associated with behavioral alterations, both in animals and humans [68,69,70,71]. As an example, Hill et al. [68] reported that children with sleep disorders have higher cerebral blood flow velocity than control children. In addition, activation of stress and anxiety response pathways has been associated with changes in the vasculature of the brain in rodents [69,71]. Finally, other pathologies that affect the central nervous system, such as type 2 diabetes and Alzheimer’s disease, and those that present anxious behavior, also present dysfunction of the cerebral vasculature [70]. Therefore, and based on our results, it is likely that the anxious-like behavior profile observed in TrJ/+ mice may be linked to changes in the cerebral vasculature.

## 5. Conclusions

Central vascular involvement has been demonstrated to be a component associated with major CNS disorders. However, this involvement noted in Charcot and Marie’s early work has been subsequently scarcely explored in CMT, pointing to the involvement of the autonomic nervous system. Our work contributes to the description and elucidation of hemodynamic changes recorded in vivo, associated with vessel volume modulation. In addition, behavioral alterations of the anxious type converge in the TrJ/+ model in a functional-structural-behavioral profile that demonstrates the vascular/central involvement of the disease. Thus, the requirement for increased hippocampal blood flow in TrJ/+ could respond to increased metabolic activity with increased oxygen demand to sustain the higher levels of anxiety. Future works will be needed to confirm this hypothesis and its implications to understand more deeply the neuropathy and its therapeutic implications.

## Figures and Tables

**Figure 1 biology-12-01324-f001:**
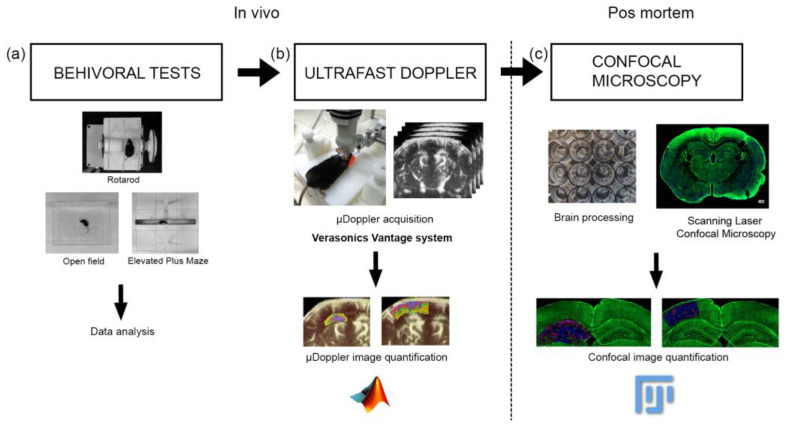
Workflow showing the experiment method. (**a**) First, the behavioral tests were performed, (**b**) then the µDoppler im-200 ages were acquired, (**c**) finally the brains were extracted and were processed for confocal microscopy visualization. The data 201 processing and the statistical analysis were performed for each experiment.

**Figure 2 biology-12-01324-f002:**
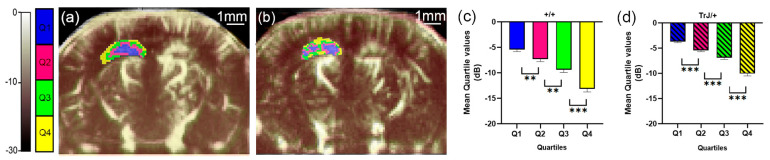
µDoppler quartile segmentation of the hippocampus. Brain coronal μDoppler images of (**a**) +/+ and (**b**) TrJ/+ mice. The color scale in dB was determined using the maximum intensity within the hippocampus as a reference. The quartile distribution of the left hippocampus is highlighted in colors. Pixels falling within the quartiles Q1, Q2, Q3 and Q4, are colored blue, fuchsia, green, and yellow, respectively. Significant differences were obtained for the mean quartile values in (**c**) +/+, and (**d**) TrJ/+ mice. ** *p* < 0.001, *** *p* < 0.0001.

**Figure 3 biology-12-01324-f003:**
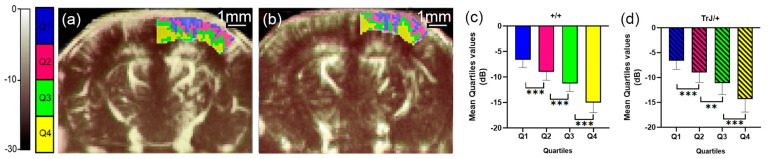
µDoppler quartile segmentation of the cortex. Brain coronal μDoppler images of (**a**) +/+ and (**b**) TrJ/+ mice. The quartile distribution is highlighted in colors on the rightcortex. Pixels falling within the quartiles Q1, Q2, Q3, and Q4, are colored blue, fuchsia, green, and yellow, respectively. Contrary to Figure 1, for this figure, the color scale in dB was computed with the maximum intensity within the cortex as reference. Significant differences were obtained for the mean quartile values in (**c**) +/+, and (**d**) TrJ/+ mice. ** *p* < 0.001, *** *p* < 0.0001.

**Figure 4 biology-12-01324-f004:**
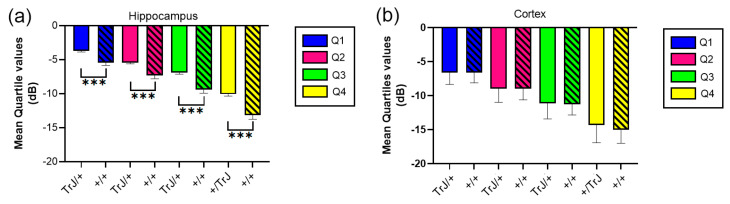
Mean quartile values of µDoppler mesure for +/+ vs. TrJ/+ mice in the hippocampus and the cortex. (**a**) Mean quartile values in the hippocampus showed significantly higher values for TrJ/+ mice when compared to +/+ mice. (**b**) No significant differences were found in the cortex. *** *p* < 0.0001.

**Figure 5 biology-12-01324-f005:**
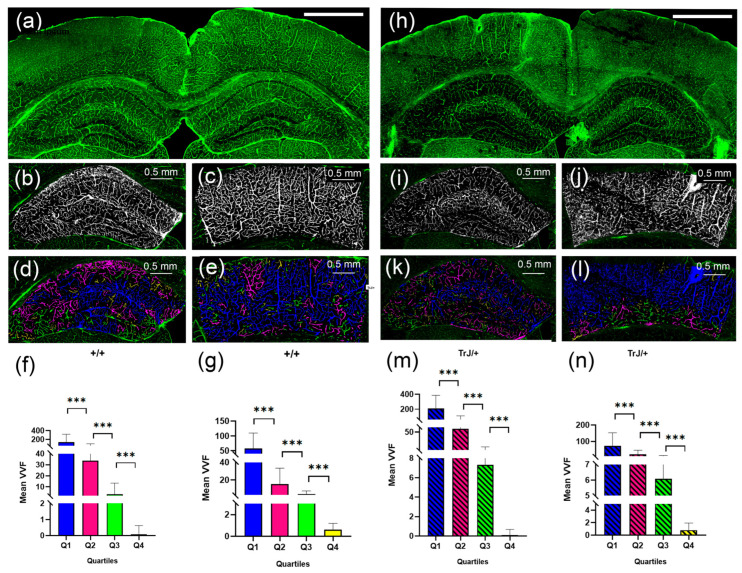
Vascular structure by SLCM. (**a**) Tile-scan image of a coronal section from a +/+ mouse brain. (**b**) Binary image in black and white by Image J automatic Threshold from the hippocampus in (**a**). (**c**) Binary image in black and white by Image J automatic Threshold from the cortex section in (**a**). (**d**) 3D counter object image showing the distribution of identified vessels. Vessels in the Q1, Q2, Q3, Q4 range were colored blue, fuchsia, green, and yellow, respectively. (**e**) same as (**d**) but for the cortex section. (**h**–**l**) same as (**a**–**e**) but for TrJ/+ mouse. (**f**,**g**,**m**,**n**) show the mean vessel volume fraction (VVF) in the hippocampus and the cortex section, for all +/+ and TrJ/+ mice included in the study, respectively. All quartiles show significant differences. *** *p* < 0.0001. The white bar in (**a**) and (**h**) represent 1 mm.

**Figure 6 biology-12-01324-f006:**
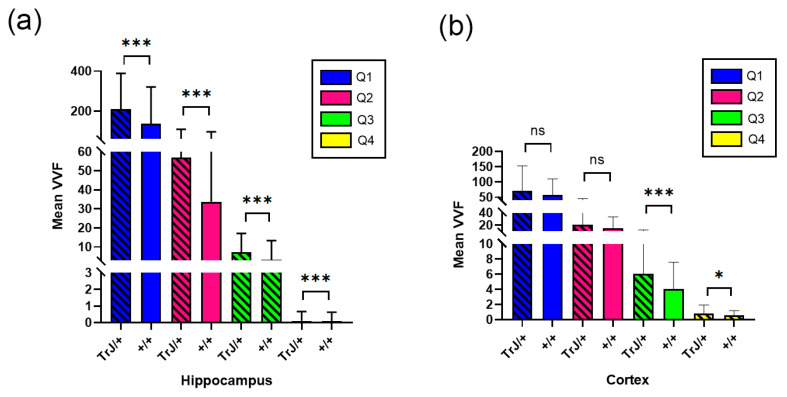
VVF Mean values for +/+ and TrJ/+ mice. (**a**) Comparison obtained in the hippocampus. IB4 probing, tile-scan SLCM-imaging and 3D Counter FIJI plug-in showed significant differences for all quartiles in the VVF values in the hippocampus between +/+ and TrJ/+ mice,. (**b**) Same as (**a**) but in the cortex section. No significant differences were found for Q1 and Q2; Q3 and Q4 showed significant differences. * *p* < 0.01, *** *p* < 0.0001, ns not significant.

**Figure 7 biology-12-01324-f007:**
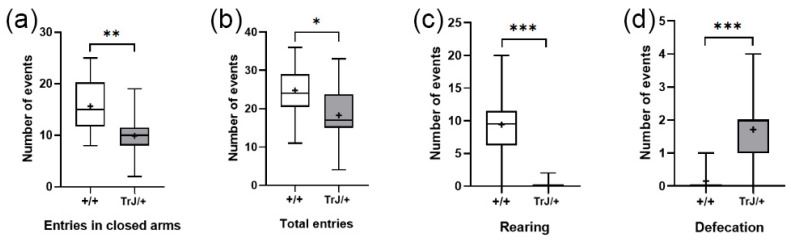
Behavioral parameters of +/+ and TrJ/+ mice in the Plus Maze test. (**a**) Entries in closed arms. (**b**) Total entries. (**c**) Rearing. (**d**) Defecation. Parameters in (**c**,**d**) were not normally distributed and were analyzed using the Mann–Whitney U-test. Parameters in (**a**,**b**) were normally distributed and analyzed using Student’s *t*-test. * *p* < 0.01, ** *p* < 0.001, *** *p* < 0.0001.

**Figure 8 biology-12-01324-f008:**
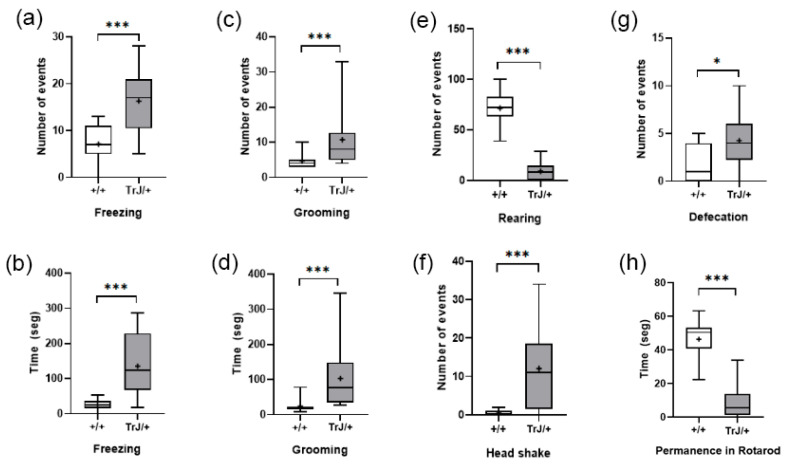
Behavioral parameters of +/+ and TrJ/+ mice in the open field and Rotarod test. (**a**) Frequency of freezing. (**b**) Time spent in freezing. (**c**) Frequency of grooming. (**d**) Time spent grooming. (**e**) Frequency of rearing. (**f**) Frequency of head shakes. (**g**) Defecation. (**h**) Time of permanence in the Rotarod test. Only (**b**) was distributed normally and was analyzed using Student’s *t*-test. The rest of the parameters were analyzed using the Mann–Whitney U-test. * *p* < 0.01, *** *p* < 0.0001.

**Table 1 biology-12-01324-t001:** Distribution of animals used for each test.

Genotype	Behavioral Tests	µDoppler	Confocal Microscopy
+/+	12	12	4
TrJ/+	12	12	4

## Data Availability

Data available on request due to restrictions e.g., privacy or ethical. The data presented in this study are available on request from the corresponding author.

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
