# Peer review of "In Vivo Ultrafast Doppler Imaging Combined with Confocal Microscopy and Behavioral Approaches to Gain Insight into the Central Expression of Peripheral Neuropathy in Trembler-J Mice"

_biology, 2023, doi:10.3390/biology12101324_

Round 1
Reviewer 1 Report
My suggestions:
1. In the introduction, I would add a few examples of genes, involved in CMT.
2. I would describe the gene and protein structure of PMP22 in a little bit of detail. Were there any other pathogenic mutations described on PMP22 besides L16P?
3. For Chapter 2.1 I would add a table on mice, involved in the study. For example, how many homo-and heterozygous mice were used in the experiments? and how many were used for the different tests?
4. In the Methods section, I would add a workflow of the experiment.
5. Did the authors test TrJ/TrJ mice and compared them with TrJ/+ and +/+ mice? If no, it may be interesting to discuss, whether the homozygous genotype would result in more aggressive phenotypes.
6. In the Discussion I would add a theoretical pathway on the possible pathogenic mechanisms of TrJ/+ mice.
7. Is it possible that the pathogenic mechanisms of TrJ/+ mice may be associated with Notch signaling due to the vascular involvement?
Reviewer 2 Report
This is a very informative article introducing a well-designed study that correlates behaviors, e.g., anxiety, with physiology and underlying biological changes in CMT. I only have two following suggestions:
1. Instead of focusing on what methods were used in the research, I suggest briefly describing the study conclusion in the article title and Simple Summary.
2. In the Introduction section, I recommend reviewing the treatment options for CMT. In the Discussion section, I recommend that the authors examine and hypothesize what this finding could mean for therapeutic options for CMT.
The English language in this article is generally understandable. There are misplaced commas and indefinite articles, e.g., "a", along with other grammatical errors. I recommend thorough reviewing of the manuscript to correct these errors and improve the quality of the writing.
Round 2
Reviewer 1 Report
Thank you, reviewers fulfilled my suggestions.